

# High expression of stromal signatures correlated with macrophage infiltration, angiogenesis and poor prognosis in glioma microenvironment

Yixin Tian[1,2], Yiquan Ke[1,2] and Yanxia Ma[1,2]

[1] Department of Neurosurgery, Zhujiang Hospital, Southern Medical University, The National Key Clinical Specialty, The Engineering Technology Research Center of Education Ministry of China, Guangdong Provincial Key Laboratory on Brain Function Repair and Regeneration, The Neurosurgery Institute of Guangdong Province, Guangzhou 510282, China
[2] Key Laboratory of Mental Health of the Ministry of Education, Guangdong-Hong Kong-Macao Greater Bay Area Center for Brain Science and Brain-Inspired Intelligence, Southern Medical University, Guangzhou 510515, China

Corresponding authors
Yiquan Ke, kyquan@smu.edu.cn
Yanxia Ma, laoma888xin@sina.com

## ABSTRACT

Glioma is one of the most fatal tumors in central nervous system. Previous studies gradually revealed the association between tumor microenvironment and the prognosis of gliomas patients. However, the correlation between tumor-infiltrating immune cell and stromal signatures are unknown. In our study, we obtained gliomas samples from the Chinese Glioma Genome Atlas (CGGA) and The Cancer Genome Atlas (TCGA). The landscape of tumor infiltrating immune cell subtypes in gliomas was calculated by CIBERSORT. As a result, we found high infiltration of macrophages was correlated with poor outcome ($P < 0.05$). Then functional enrichment analysis of high/low macrophage-infiltrating groups was performed by GSEA. The results showed three gene sets includes 102 core genes about angiogenesis were detected in high macrophage-infiltrating group. Next, we constructed PPI network and analyzed prognostic value of 102 core genes. We found that five stromal signatures indicated poor prognosis which including HSPG2, FOXF1, KDR, COL3A1, SRPX2 ($P < 0.05$). Five stromal signatures were adopted to construct a classifier. The classifier showed powerful predictive ability (AUC = 0.748). Patients with a high risk score showed poor survival. Finally, we validated this classifier in TCGA and the result was consistent with CGGA. Our investigation of tumor microenvironment in gliomas may stimulate the new strategy in immunotherapy. Five stromal signature correlated with poor prognosis also provide a strong predator of gliomas patient outcome.

## INTRODUCTION

Glioma is the most deadly central nervous system (CNS) tumor. Patients with gliomas always show poor outcome, and their median overall survival remains from 14.6 to 16.8 months. The prognosis of gliomas patients is correlated with tumor subtype, age and sex (*Mortazavi, Mortazavi & Paknahad, 2018*). Based on tumor morphology and

molecular alterations, the WHO updated a new classification system in 2016 (*Molinaro et al., 2019*). Gliomas are classified based on the absence or presence of mutations in IDH and the absence or presence of 1p/19q chromosomal co-deletion (*Lim et al., 2018*): IDH-mutant, 1p/19q-co-deleted present a benign prognosis; IDH-mutant, non-1p/19q-codeleted have intermediate survival outcomes; and IDH-wild-type have an unfavorable prognosis. The poor prognosis of gliomas has stimulated the discovery of new treatment strategies such as immunotherapy (*Gan et al., 2017*; *Broekman et al., 2018*). However, gliomas present significant resistance to immunotherapy. Resistance of innate immunity prevents the tumor from an immune response, resistance of adaptive immunity deactivates tumor-infiltrating immune cells (*Jackson, Choi & Lim, 2019*). Thus, in order to carry out further study of immunotherapy, it is vital to figure out the tumor microenvironment in gliomas. Previous study integrated several tumor microenvironment related genes based on The Cancer Genome Atlas (TCGA) database applied by ESTIMATE algorithmbased immune scores (*Chen et al., 2018*). Even so, the correlation between tumor-infiltrating immune cells and stromal signatures in gliomas are still unknown.

The Chinese Glioma Genome Atlas (CGGA) is a database which contains 2,000 glioma samples from Chinese cohorts. This database was finished a large scale update in July 2019, and 693 mRNA sequencings and matched clinical data were provided, which includes: histology, WHO grades, age, gender, chemotherapy, radiotherapy, overall survival. CIBERSORT is a computational method for quantifying 22 immune cell subtypes fractions from gene expression profile such as microarray or RNA-seq (*Chen et al., 2018*). Flow cytometry verified this computational method and several studies showed the landscape of tumor-infiltrating immune cells can be accurately determined by this method in various malignant tumors such as colon cancer, breast cancer, lung cancer and prostate cancer (*Zhang et al., 2019*; *Ali et al., 2016*; *Liu et al., 2017*).

In this study, we extracted 693 RNA-seq and matched clinical data from the CGGA database. Then, we analyzed the differences of 22 immune cell subtypes between normal brain tissue and gliomas, and revealed their prognosis value respectively. We also compared immune cell subtypes among different phenotypes such as: histology, WHO grades, age, gender, chemotherapy, radiotherapy, IDH and 1p/19q chromosomal co-deletion. Next, we identified five stromal signatures with high macrophages infiltration and angiogenesis which significantly contribute to prognosis by functional enrichment analysis and protein–protein interaction networks. And the stromal signatures were adopted to construct a classifier. The classifier showed powerful predictive ability (AUC = 0.748). Patients with a high risk score (RS) showed poor survival. Finally, we validated this classifier in the TCGA database and the result was consistent with the CGGA database. The results provide a comprehensive view of the tumor microenvironment and may stimulate new strategies of immunotherapy in Chinese glioma patients.

## METHODS

### Collecting RNA-seq and matched clinical data from CGGA and TCGA

We built two independent cohorts for training set and testing set from CGGA and TCGA. The gene expression profiles were downloaded from The Chinese Glioma Genome Atlas

**Table 1 Clinical features of glioma patients in CGGA and TCGA.**

|  | CGGA ($n$ = 693) | TCGA ($n$ = 668) |
| --- | --- | --- |
| Age (year) | | |
| >45 | 276 (39.8%) | 337 (50.4%) |
| ≤45 | 416 (60.0%) | 331 (49.6%) |
| Gender | | |
| Male | 398 (57.4%) | 385 (57.6%) |
| Female | 295 (42.6%) | 283 (42.4%) |
| WHO grade | | |
| WHO II | 188 (27.1%) | 247 (37.0%) |
| WHO III | 255 (36.8%) | 261 (39.1%) |
| WHO IV | 249 (35.9%) | 0 (0.0%) |
| OS (year) | | |
| ≤1 | 163 (23.5%) | 244 (36.5%) |
| 1–3 | 236 (34.1%) | 298 (44.6%) |
| 3–5 | 111 (16.0%) | 70 (10.5%) |
| ≥5 | 110 (15.9%) | 56 (8.4%) |

(CGGA, http://www.cgga.org.cn/). A total of 695 RNA-seq data of Chinese gliomas patients were obtained from the CGGA database. A series of measures were taken: (1) Genes with a variance of 0 will be filtered out. (2) Complete follow-up information and samples with a follow-up time >30 days will be served. Finally, 693 samples meeting the inclusion criteria were included. And each patient matched specific clinical data, which includes: histology, WHO grades, age, gender, chemotherapy, radiotherapy, survival status, survival duration in days. To build the testing cohort, The RNA-seq FPKM of gliomas including corresponding outcome data were downloaded from The Cancer Genome Atlas (TCGA, https://cancergenome.nih.gov/). The following measures were taken: (1) The IDs were annotated on the basis of hg38 reference genome. (2) Genes with a variance of 0 will be filtered out. (3) For the same gene corresponding to multiple IDs or a patient with multiple tumor samples, the average will be taken. (4) Samples with a follow-up time >30 days were remained. Eventually, a total of 668 TCGA samples fulfilled our criteria. The details of patients information was showed in Table 1.

## Analysis of tumor-infiltrating immune cells

CIBERSORT is a computational method for quantifying immune cell subtypes fractions from normalized gene expression profiles based on deconvolution algorithm. The leukocyte gene signature matrix (LM22) is generated for the calculation of 22 human hematopoietic subsets. LM22 has been verified on Affymetrix HGU133 and Illumina Beadchip platforms. In our study, we used LM22 gene file to calculate 22 immune cells subtypes of 693 gliomas patients. These immune cell subtypes included CD8+ T cells, naïve CD4+ T cells, resting CD4+ memory T cells, activated CD4+ memory T cells, follicular helper T cells (Tfh), Tregs, γδ T cells, naïve B cells, memory B cells, plasma cells, resting NK cells, activated NK cells, resting NK cells, activated NK cells, resting

dendritic cells, activated dendritic cells, Monocytes, Eosinophils, Neutrophils M0 macrophages, M1 macrophages, M2 macrophages, resting mast cells and activated mast cells. And CIBERSORT $p$-value < 0.05 was demanded. The median cutoff was determined to separate the patients into the high-infiltrating or low-infiltrating groups.

## Statistical analysis

The differences of immune cell subtypes between normal brain tissues and gliomas tissues were assessed using the unpaired $t$-test, and the differences of tumor-infiltrating immune cells among different phenotypes were performed by $t$-test and one One-way analysis of variance (ANOVA). Kaplan–Meier curves were used to perform the correlations of 22 immune cell subtypes or gene signature and corresponding clinical follow-ups (log-rank test), and high vs low groups all based on median cut off. Statistical analyses were calculated by SPSS statistical package software or GraphPad Prism. $P$ values < 0.05 were considered significant.

## Enrichment analysis

Functional enrichment analyses of tumor-infiltrating immune cells were performed by Gene Set Enrichment Analysis (GSEA). We analyzed via GSEA v4.0.3 for Windows (http://software.broadinstitute.org/gsea/index.jsp) (*Subramanian et al., 2005*). GO categories (*Ashburner et al., 2000*) include biological processes (BP), molecular functions (MF), or cellular components (CC) and Kyoto Encyclopedia of Genes and Genomes (KEGG) (https://www.kegg.jp/) pathway were analyzed by GSEA (*Reimand et al., 2019*). FDR < 0.1 and $P$ < 0.01 were considered significant. Protein–protein interaction (PPI) networks were constructed by the STRING tool (https://string-db.org/) and analyzed by Cytoscape (*Shannon et al., 2003*). PPI network was used to identify the hub gene. And the gene set which contain 149 stromal signatures was generated in both ESTIMATE and MSigDB.

## Construction of classifier

To generate and optimize the prognostic classifier, the multivariate Cox regression analysis was performed in CGGA and TCGA cohort. We calculated the RS of each sample based on the multivariate COX coefficient, and the low/high risk groups were defined according to the median cutoff RS. The receiver operator characteristics (ROC) curve analysis was applied to assess the classifier's ability to distinguish samples with a high or low RS, and it was draw by *survivalROC* package. The area under the curve (AUC) of the ROC curve was calculated and compared to examine the performance of the classifier in both training and testing cohorts. The median RS was determined to separate the genes into the high-risk or low-risk groups. KM curves and independent testify were performed by *survival* package to assess the effective of classifier in gliomas patients (log-rank test). All analysis were carried out by R version 3.6.1 and corresponding packages.

## Proteomics and histology

To verify the infiltration of M2 macrophage in glioma patients, we downloaded proteomics data of 110 glioma patients from The National Cancer Institute's Clinical Proteomic

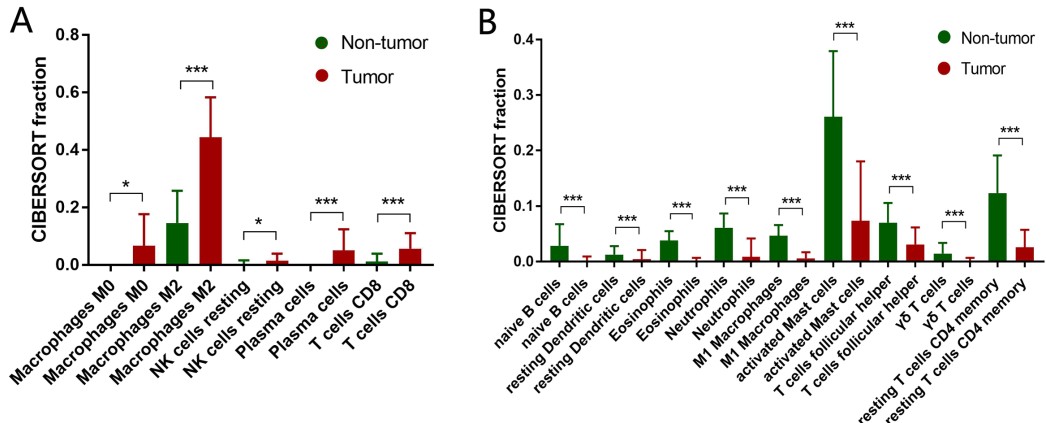

**Figure 1 Adaptive and innate immune cells in gliomas.** (A) Adaptive immune cell subtypes in glioma. The fraction of M0 macrophages, M2 macrophages, resting NK cells, plasma cells, CD8+ T cells is significantly higher in glioma tissue than normal brain tissue. (B) Innate immune cell subtypes in glioma. The fraction of naïve B cells, resting dendritic cells, Eosinophils, Neutrophils, M1 macrophages, activated mast cells, follicular helper T cells (Tfh), γδ T cells and resting CD4+ memory T cells is significantly lower in glioma tissue than normal brain tissue. *P < 0.05; **P < 0.01; ***P < 0.001.

Tumor Analysis Consortium (CPTAC; https://cptac-data-portal.georgetown.edu/). And the histological level research of glioma patients was performed in the human protein atlas (https://www.proteinatlas.org/).

# RESULT

## The landscape of tumor infiltrating immune cell subtypes in gliomas

Based on the 693 RNA-seq from CGGA database, the different infiltration of 22 immune cell subtypes between normal brain tissue and gliomas were analyzed by *t*-test. The fraction of M0 macrophages, M2 macrophages, resting NK cells, plasma cells, CD8+ T cells was significantly higher in gliomas tissue than normal brain tissue (Fig. 1A). On the contrary, the fraction of naïve B cells, resting dendritic cells, Eosinophils, Neutrophils, M1 macrophages, activated mast cells, follicular helper T cells (Tfh), γδ T cells and resting CD4+ memory T cells was significantly lower in gliomas tissue than normal brain tissue (Fig. 1B).

A total of nine clinical parameters were analyzed, which includes: age, gender, histology, WHO grades, primary or recurrence, chemotherapy, radiotherapy, mutations in IDH and 1p/19q co-deletion. As a result, gender, primary or recurrence, chemotherapy showed no significant difference of immune cells. Monocytes were decreased in elderly patients (Fig. 2A). M0 macrophages, Tregs and activated dendritic cells were increased in high grade gliomas, whereas monocytes were decreased (Fig. 2B). The fraction of M0 macrophages, Tregs and γδ T cells was higher in glioblastoma (GBM) than astrocytoma (AOA), whereas monocytes and activated mast cells was lower (Fig. 2C). The fraction of activated mast cells, monocytes and resting CD4+ memory T cells was higher in IDH mutant than wildtype, while M0 macrophages, Tregs, γδ T cells and Tfh was lower

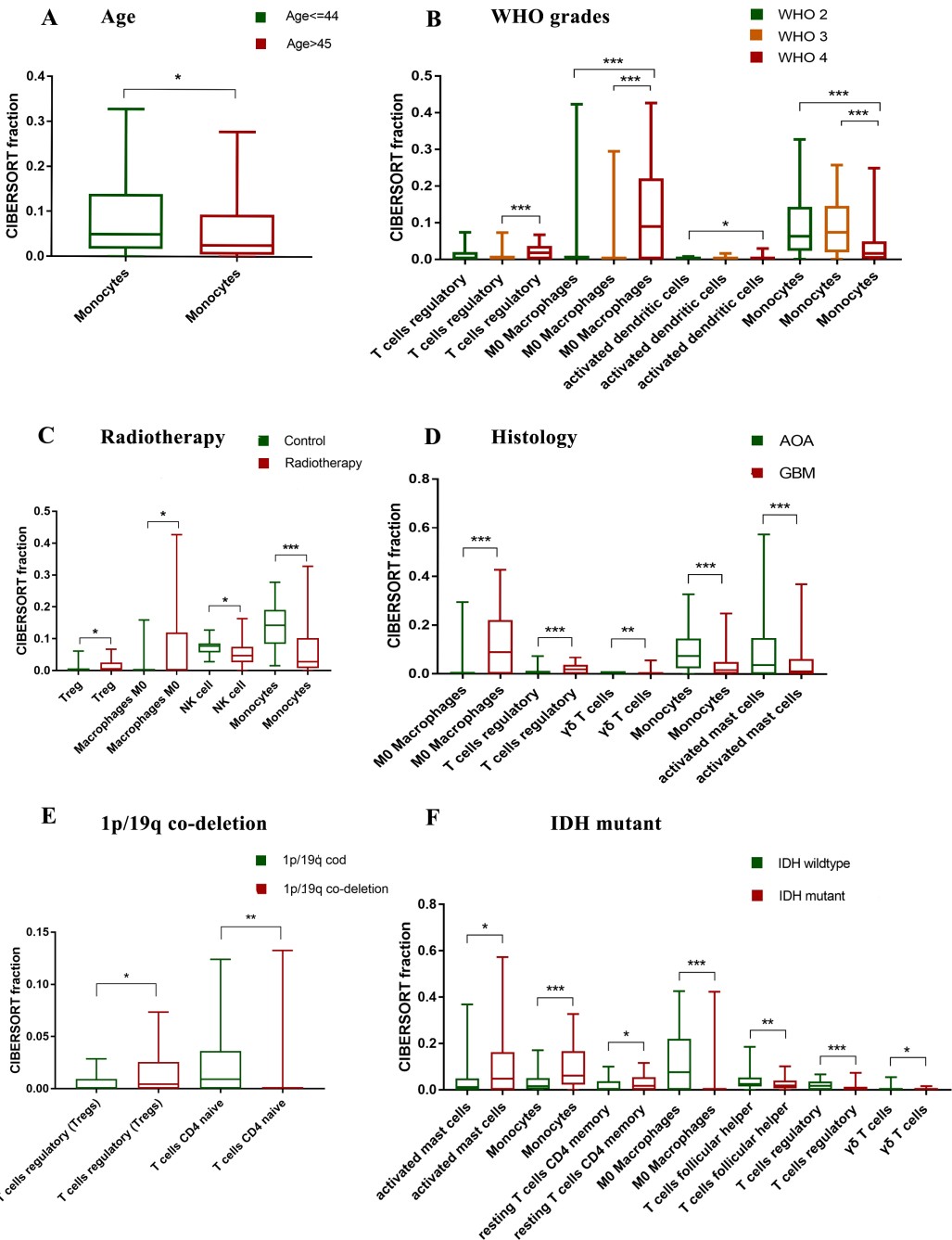

**Figure 2 Correlation between clinical parameters and immune cells.** (A) Monocytes are decreased in elder patients. (B) M0 macrophages, Tregs and activated dendritic cells are increased in high grade gliomas, whereas monocytes are decreased. (C) Tregs and M0 macrophages are increased in patients with radiotherapy, whereas NK cells and Monocytes are decreased. (D) M0 macrophages, Tregs and γδ T cells are increased in glioblastoma (GBM), whereas monocytes and activated mast cells are decreased. (E) Tregs are increased in 1p/19q co-deletion, whereas naïve CD4+ T cells are decreased. (F) Activated mast cells, monocytes and resting CD4+ memory T cells are increased in IDH mutant, whereas M0 macrophages, Tregs, γδ T cells and Tfh are decreased. $^*p < 0.05$; $^{**}p < 0.01$; $^{***}p < 0.001$.

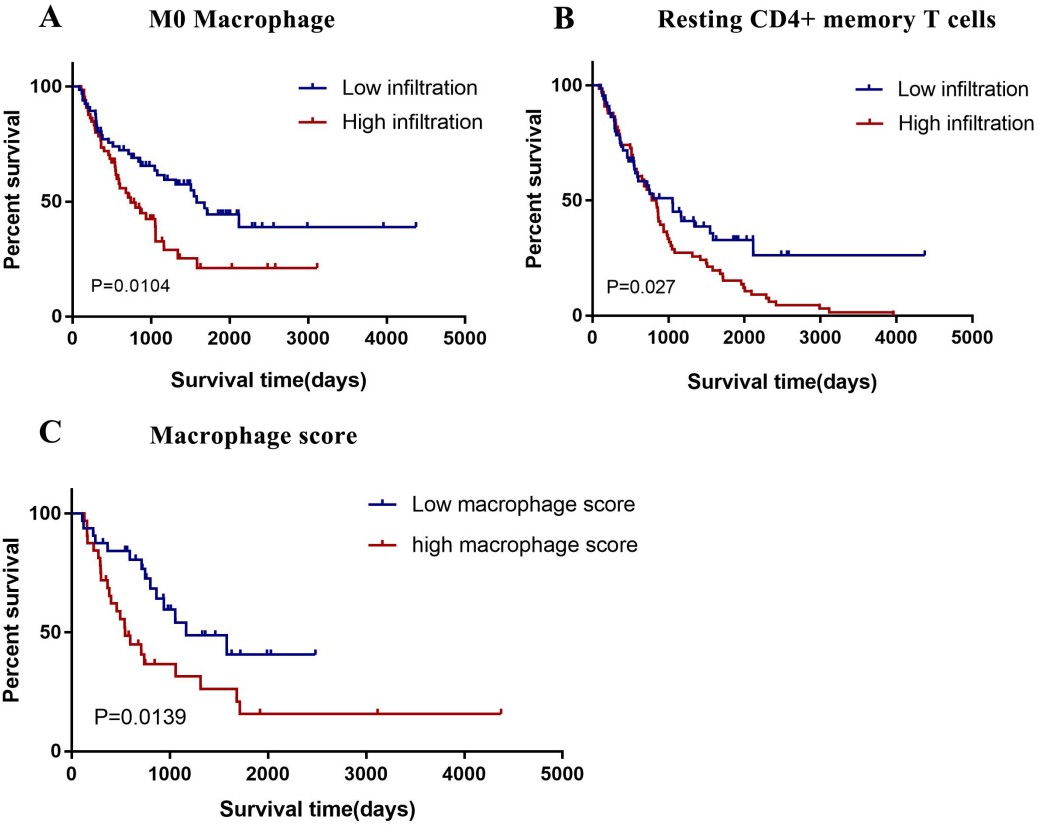

**Figure 3** **Prognostic value of tumor-infiltrating immune cells in gliomas.** (A) The high infiltration of resting CD4+ memory T cells are correlated with poor outcome ($P$ = 0.027). (B) The high infiltration of M0 macrophages are correlated with poor outcome ($P$ = 0.0104). (C) High macrophage score are correlated with poor outcome ($P$ = 0.0139).

(Fig. 2D). Tregs were increased in 1p/19q co-deletion, whereas naïve CD4+ T cells were decreased (Fig. 2E).

## Prognostic value of tumor-infiltrating immune cells in gliomas

In this study, we divided all 22 immune cell subtypes into high/low infiltration group based on median cutoff. And we analyzed the correlation between 22 immune cell subtypes and patients prognosis by generating Kaplan–Meier survival curve. Result showed that high infiltration of M0 macrophages ($P$ = 0.0104) and resting CD4+ memory T cells ($P$ = 0.027) had an unfavorable prognosis. (Figs. 3A and 3B) Based on the previous studies, M2 macrophages showed pro-tumor function as well as M1 macrophages showed anti-tumor function in gliomas. We constructed a macrophage score: (M0 + M2)/M1. And results showed gliomas patients with high macrophage score correlated with poor outcome. (Fig. 3C).

## Functional enrichment analysis of gene signature of macrophages

As the infiltration of macrophages play a significant role in glioma patients' outcomes, we further investigated the potential mechanism and gene signature. First, we divided gliomas patients into high/low macrophage-infiltrating groups. Functional enrichment analysis was

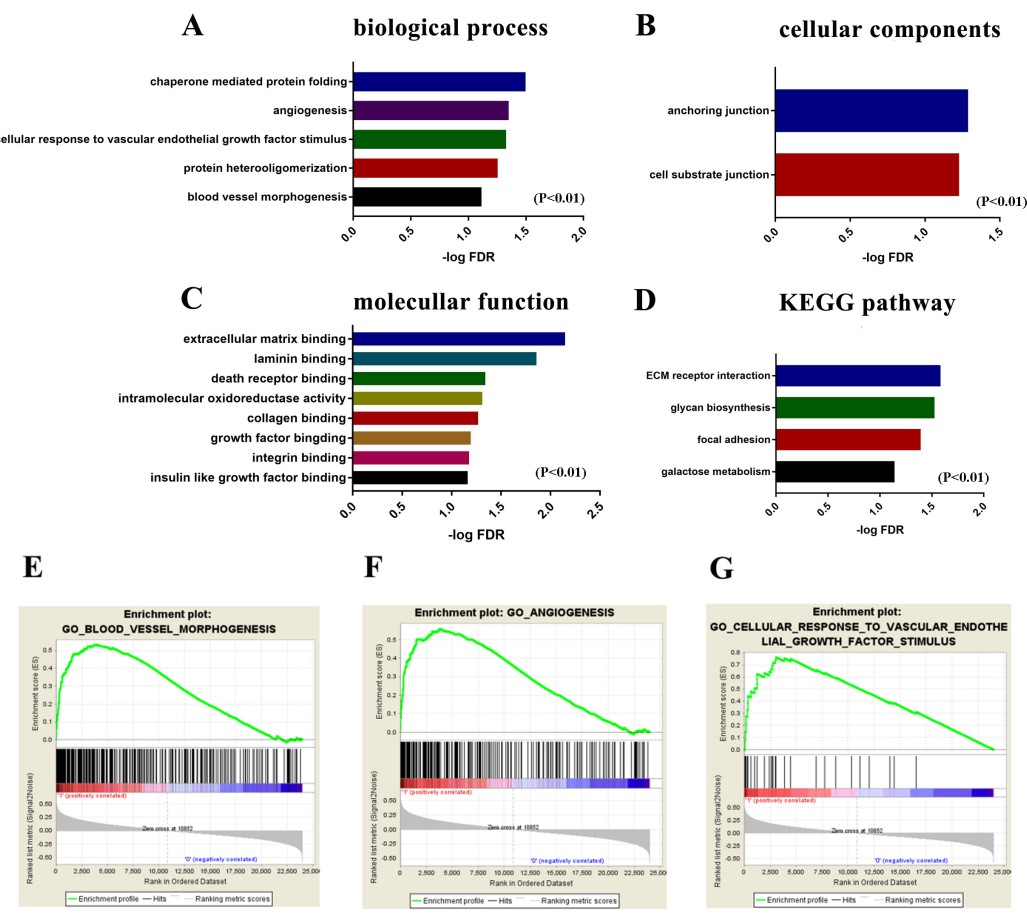

**Figure 4 Functional enrichment analysis of gene signature of macrophages.** GO term and KEGG pathway analysis for the high macrophage-infiltrating group. Pathways with FDR < 0.1 $P$ < 0.01 are shown. (A) Biological process. (B) Cellular components. (C) Molecular function. (D) KEGG pathway. (E) Enrichment profile generated with GSEA in the gene set "blood vessel morphogenesis" comparing high vs. low M0 macrophage-infiltrating group. (F) Enrichment profile generated with GSEA in the gene set "angiogenesis". (G) Enrichment profile generated with GSEA in the gene set "cellular response to vascular endothelial growth factor stimulus".

performed by Gene Set Enrichment Analysis Gene Set Enrichment Analysis (GSEA). GO categories include BP, MF, CC and Kyoto Encyclopedia of Genes and Genomes (KEGG) pathway were analyzed. As a result, a total of 5 GO terms of biological process, 2 GO terms of cellular component, 8 GO terms of molecular function and 4 KEGG pathway terms were identified to be significant (FDR < 0.1 and $P$ < 0.001) (Fig. 4). Interestingly, the 3 enrichment GO terms with high macrophage infiltration were correlated with angiogenesis, so we obtained the 102 core gene signatures for next investigation.

## Investigation of stromal signatures correlated with macrophage infiltration and angiogenesis

We built PPI networks to further investigate the 102 core gene signatures which correlated with high macrophage infiltration and angiogenesis by STRING tool. Cytoscape and Cytohubba were used to analyze hub genes (Fig. S3). And the gene set which contain

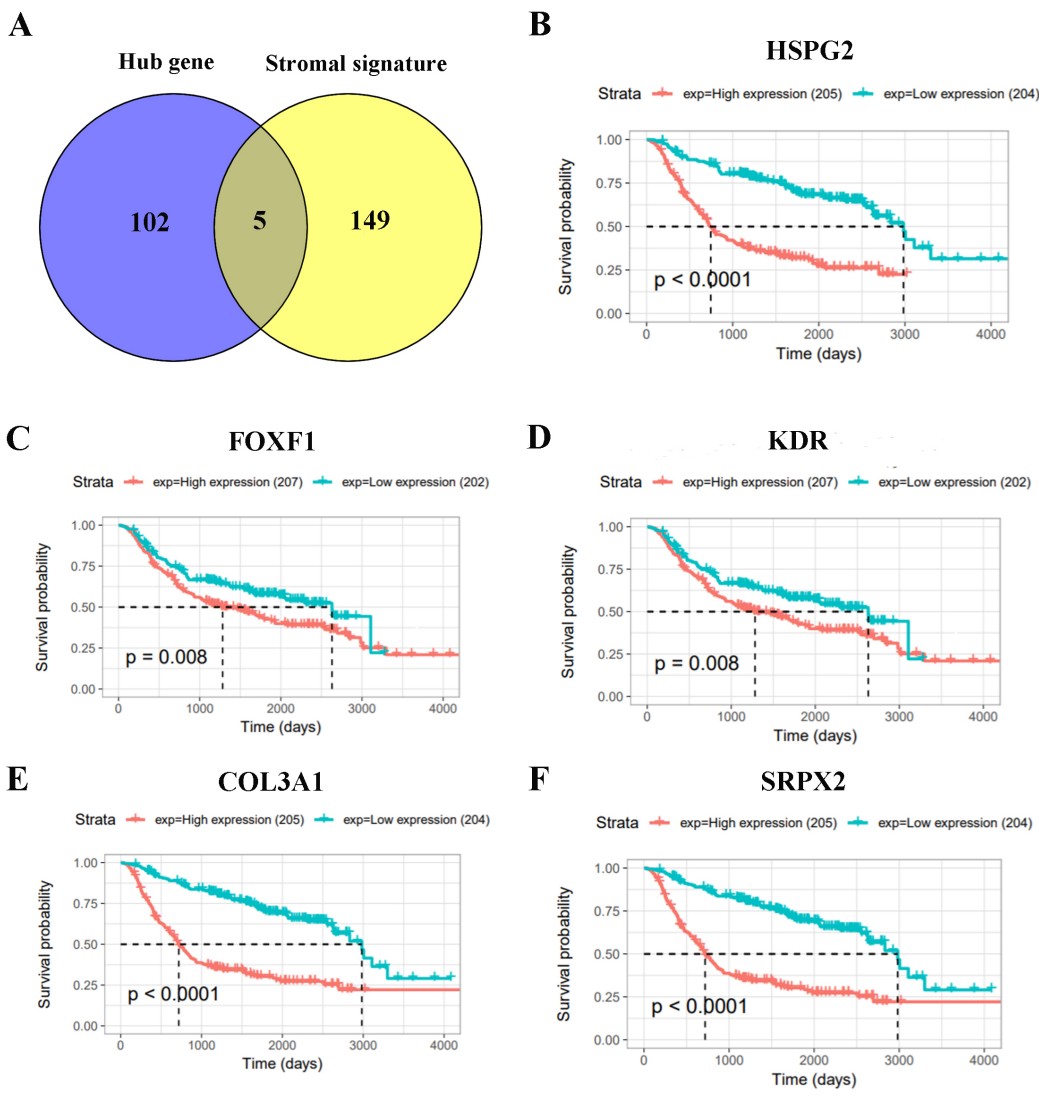

**Figure 5 Investigation of prognostic value of stromal signatures.** (A) Venn diagrams showing the number of commonly genes in hub genes group and stromal signatures. (B–F) The prognostic value of five stromal signatures by Kaplan–Meier survival curves in the CGGA database.

149 stromal signatures was provided by ESTIMATE and MSigDB. Then, we compared 102 hub genes with 149 stromal signatures. Eventually we generated five stromal signatures including HSPG2, FOXF1, KDR, COL3A1, SRPX2 (Fig. 5A). To investigate the prognostic value of five stromal signatures, we analyzed overall survival by Kaplan–Meier survival curves in the CGGA database. The result show that five stromal signatures were significantly correlated with poor outcome ($P < 0.05$) (Figs. 5B–5F).

## Construction and verification of prognostic classifier based on stromal signatures

According to univariate COX regressions, all of five stromal signatures were significantly correlated with poor outcome in gliomas patients (HR > 1) (Fig. 6A). We calculated the RS

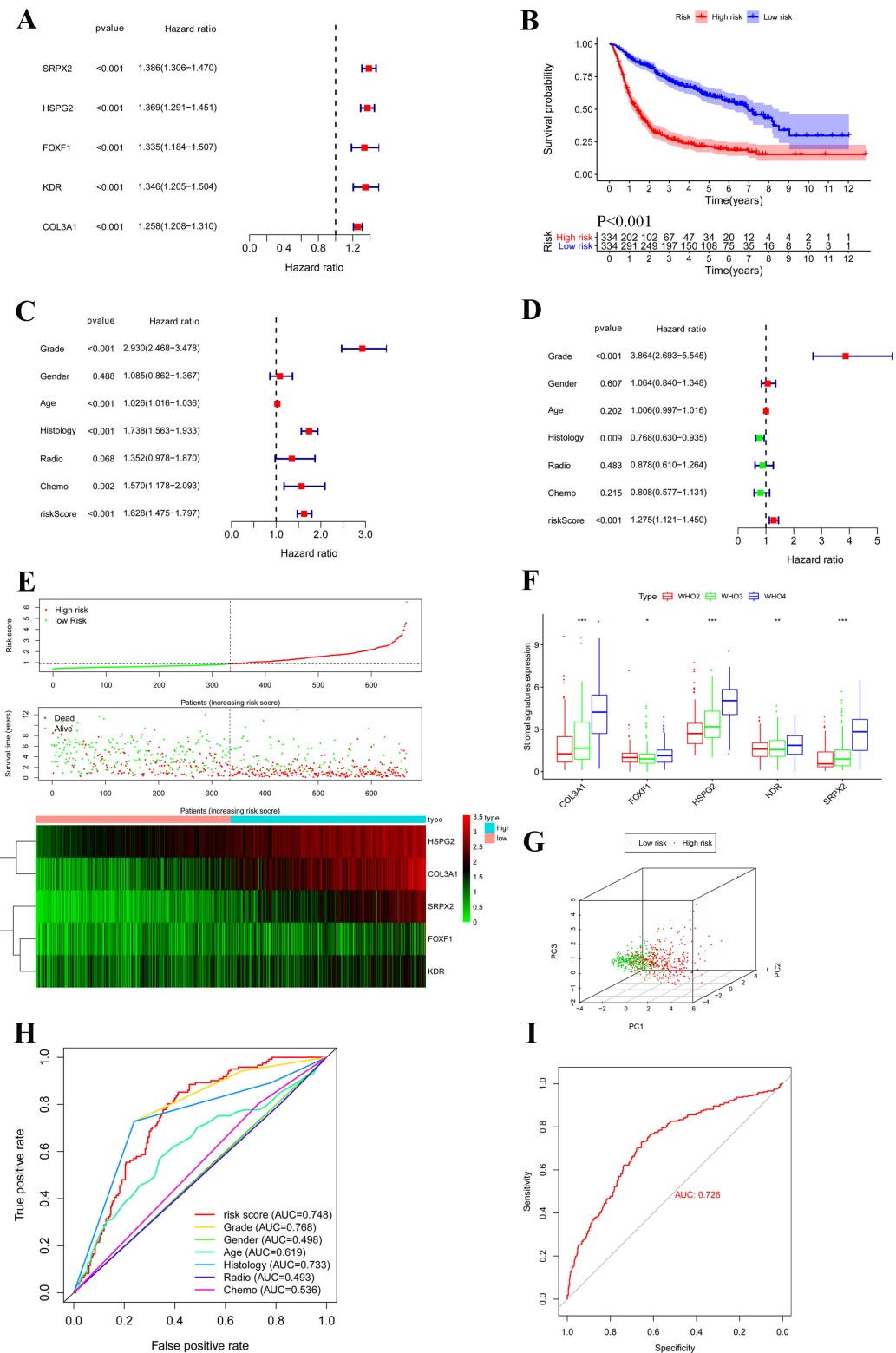

**Figure 6 Construction and verification of prognostic classifier based on stromal signatures.** (A) Forest map based on the univariate COX regressions of five stromal signatures. Right vertical dotted line indicates risk genes. (B) Kaplan–Meier survival curves were generated for the comparison of groups of high (red line) and low (blue line) risk score. (C and D) Forest map based on the univariate and

**Figure 6 (continued)**
multivariate COX regressions which enrolled clinical features and stromal signatures in the overall set. (E) Five stromal signatures were enrolled in the risk model heatmap. (F) Boxplot shows five stromal signatures were correlated with WHO grades. (G) Principal component analysis shows significant different between groups of low (green dots) and high (red dots) risk score. (H) Receiver operating characteristic (ROC) curve plotted to determine the effect of the classifier and compare with other clinical features. (I) The classifier was verified in an independent TCGA cohort. $^*p < 0.05$; $^{**}p < 0.01$; $^{***}p < 0.001$.

of each sample based on the multivariate COX coefficient, and the low/high risk groups were defined according to the median cutoff RS. Then, K–M plot showed patients with high RS had lower survival than patient with low RS ($P < 0.001$) (Figs. 6B and 6E). Next, we generated univariate and multivariate analyses which enrolled clinical features and stromal signatures in the overall set. The result showed that stromal signature classifier was an independent factor for gliomas patients (Figs. 6C and 6D). Furthermore, we found all of five stromal signatures were correlated with WHO grades ($P < 0.05$) (Fig. 6F). As for histology, the expression of stromal signatures in GBM was significantly higher than AOA and oilgodendroglioma. The expression of COL3A1, HSPG2 and SRPX2 in patients with chemotherapy was higher than patients without chemotherapy. And the expression of COL3A1, KDR and SRPX2 was higher than patients with radiotherapy than patient without radiotherapy (Fig. S1). Principal component analysis showed significant different between groups of low and high RS (Fig. 6G). A receiver operating characteristic (ROC) curve was plotted to assess the effect of the classifier. The AUC of the classifier was 0.748. Although stromal classifier was less efficient than the WHO grade, it was much superior to histological type (Fig. 6H). To verify the classifier, we generated an independent cohort based on TCGA dataset. And the classifier also showed strong predictive ability in TCGA cohort (AUC = 0.726) (Fig. 6I).

## Verifying the infiltration of M2 macrophage in glioma patients

CD163 is the marker protein of M2 macrophage. By measuring the expression of CD163, the infiltration degree of M2 macrophage in glioma can be determined. We demonstrated high expression of CD163 in glioma patients through the CPTAC proteomics database. Meanwhile, we compared the immunohistochemistry of CD163 in normal brain tissue and glioma through the human protein atlas, and found that the expression of CD163 in glioma was significantly higher than that in normal brain tissue (Fig. S2).

## DISCUSSION

First of all, based on the RNA-seq data from CGGA and the CIBERSORT, we calculated 22 subtypes of tumor-infiltrating immune cells in gliomas. We comprehensively analyzed the tumor-infiltrating immune cells present in gliomas and revealed the prognosis value.

The landscape of tumor-infiltrating immune cells in gliomas as follows:

1. B cell lines: Naïve B cells can differentiate into antibody-secreting plasma cells after being stimulated by antigen (*Liao et al., 2016*). Our study found that naïve B cell was a

significant decrease in gliomas, meanwhile, the plasma cells were significant increased. It was illustrated that naïve B cell may be differentiated into plasma cells by gliomas antigens.

2. T cell lines: Tregs are correlated with unfavorable prognosis in the several kinds of tumor microenvironment (e.g., ovarian cancer, breast cancer, kidney cancer and pancreatic cancer) (*Mirzaei, Sarkar & Yong, 2017*). Our results showed that Tregs were significantly increased in GBM and high WHO grade, which demonstrated Tregs may promote the development of gliomas (*Banissi et al., 2009*; *Vandenberk & Van Gool, 2012*). Previous study showed that tumor-infiltrating Tfh cells have protective roles by suppressing lymphoid tumor-promoting effects in breast cancer and colorectal cancer (*Woroniecka et al., 2018*). Our results also shown that Tfh cells was significantly decrease in gliomas. Previous study showed neutrophils and eosinophils are implicated in almost every stage of oncogenesis and display both anti- and pro-tumor properties (*Curran, Evans & Bertics, 2011*; *Schneider, Kwan & Boockvar, 2018*).

3. Myeloid cell lines: Our results showed that neutrophils and eosinophils were significantly decreased in gliomas, which indicated neutrophils and eosinophils may have negative correlation with gliomas. Mast cells play an important role in the growth of tumors (*Põlajeva et al., 2011*). However, the contribution of mast cells in the microenvironment of solid malignancies remains controversial. Previous studies have illustrated that gliomas are associated with a profound accumulation of mast cells, and STAT5 play an important role on the recruitment of mast cells to gliomas (*Põlajeva et al., 2014*). Our results showed that resting mast cells were increased in gliomas, whereas activated mast cells were significantly decreased in gliomas. In addition, the infiltration of activated mast cells of GBM was prominently lower than AOA. Consistent with previous study, gliomas promoted mast cells recruitment, but gliomas may also block resting mast cell activation. And glioblstoma may show stronger ability of preventing mast cell activation than AOA. Dendritic cell vaccinations have emerged as newly strategies in the treatment of gliomas (*Garg et al., 2016*; *Wen et al., 2019*). Adjuvant dendritic cell-based immunotherapy for gliomas patients can induce long-term survival (*Li et al., 2018*). We found resting dendritic cells were significantly decreased in glioma, which was consistent with previous studies that dendritic cells recruitment correlated with favorable outcome. Previous study showed monocytes played an important role in cancer development and progression (*Põlajeva et al., 2011*). And different monocytes displayed both anti- and pro-tumor function (*Prosniak et al., 2013*). Our results showed monocytes were significant decreased in high grade glioma and GBM, which indicated stimulating monocytes recruitment may prevent progression of high grade gliomas. What's more, our results also showed monocytes were significantly lower in patients with radiotherapy than patients without radiotherapy, which may contribute to the resistance of gliomas radiotherapy.

Next, we further investigated tumor-infiltrating macrophages in gliomas. Based on their function, M0 macrophages can be differentiated into two categories: classically activated macrophages (M1 macrophages) and alternatively activated macrophages

(M2 macrophages) by stimulation of various cytokines (*Chen et al., 2019*; *Jiang et al., 2019*; *Lailler et al., 2019*). Previous studies show gliomas contain a plenty of macrophages, and tumor-associated macrophages correlate with progression and angiogenesis of gliomas by several ways, including: microRNA-macrophage feedback loop (*Bao & Li, 2019*; *Liu et al., 2019*), extracellular lipid loading (*Offer et al., 2019*) and cytokines like OPN, IRGM, IL-6 (*Hernandez-SanMiguel et al., 2019*; *Wang et al., 2018*; *Xu et al., 2019*). In addition, previous study successfully reeducated the pro-tumor M2 macrophages toward anti-tumor M1 macrophages by a dual-targeting biomimetic treatment strategy which provides a good method for the pharmacotherapy of gliomas (*Zhao et al., 2018*). Our results showed that M0 and M2 were significantly increased in gliomas, while M1 were significant decreased. The results confirmed pro-tumor function of M0 and M2 as well as anti-tumor function of M1 in gliomas. And the infiltration of M0 macrophages showed correlated with decreased survival ($P < 0.05$). In addition, we found M0 macrophages were significantly lower in IDH mutant than IDH wildtype. This result may explain glioma patients with IDH-mutant always have favorable survival outcomes. After functional enrichment analysis, 3 gene sets includes 102 genes about angiogenesis were detected in high macrophage-infiltrating group, which was consistent with previous study.

Among 102 core genes, we found five stromal signatures correlated with high macrophages infiltration and angiogenesis indicated poor prognosis ($P < 0.05$). The five stromal signatures including HSPG2, FOXF1, KDR, COL3A1 and SRPX2. *Sun et al. (2019)* constructed the multicellular gene network between gliomas and macrophages, they found macrophage-related gene signature had good prognostic value for predicting resistance to targeted therapeutics and survival of glioma patients. We found the relationship between stromal signatures and macorphages signatures in gliomas. Based on precious studies, high expression of perlecan/HSPG2 in gliomas lead to tumor promotion through the transformation of brain extracellular matrix into tumor microenvironment (*Hu et al., 2016*). In addition, high expression of COL3A1 contributes to glioma cells proliferation and migration (*Shin et al., 2017*). And up-regulation of KDR promotes proangiogenic myeloid cells, result in low-grade to high-grade transition (*Huang et al., 2017*). Moreover, glioma with high tumor associated macrophages has increased expression of SRPX2 (*Hung et al., 2016*). We also found these five stromal signatures were correlated with clinical features like grades, histology, chemotherapy and radiotherapy. The expression of stromal signatures was significantly higher in patients with higher grade and worse pathological features like GBM. Then, we constructed a prognosis classifier based on HSPG2, FOXF1, KDR, COL3A1 and SRPX2. And the classifier showed strong predictive ability in both CGGA cohort and TCGA cohort. We confirmed the high infiltration of M2 macrophages in protein and histology level via CPTAC and the human protein atlas. Eventually, based on our investigation, up-regulation of five stromal signatures may lead to macrophage infiltration and angiogenesis in gliomas patients. And our classifier also provides strong predictive ability in prognosis. Our study may provide a novel immunotherapeutic strategy and a more clearly overview of tumor microenvironment in gliomas.

However, there still exited several limitations in our research. First, the calculation results from public database may show bias. Although we have verified the results in two independent cohorts, we should continue to deeper research. Second, the macrophage infiltration and angiogenesis function of five stromal signatures in gliomas need to further confirmation in vitro and in vivo.

## CONCLUSIONS

Based on CGGA RNA-seq datasets, the landscape of 22 tumor-infiltrating immune cells in gliomas was analyzed by CIBERSORT. High infiltration of macrophages was correlated with poor outcome. After functional enrichment analysis, three gene sets includes 102 core genes about angiogenesis were detected in high macrophage-infiltrating group. Next, we found five stromal signatures indicated poor prognosis which including HSPG2, FOXF1, KDR, COL3A1, SRPX2. Five stromal signatures were adopted to construct a classifier. The classifier showed powerful predictive ability. Finally, we validated this classifier in TCGA and the result was consistent with CGGA. Our investigation of tumor microenvironment in gliomas may stimulate the new strategy in immunotherapy. The five stromal signatures correlated with poor prognosis also provide a strong predator of glioma patient outcomes.

## ACKNOWLEDGEMENTS

We thank Dr. Wang and Dr. Chen from Department of Neurosurgery at Zhujiang Hospital, Southern Medical University for providing us with useful advice.

### Funding

This work was supported by the Natural Science Foundation of China, No. 81772651, and Natural Science Foundation of Guangdong Province, No. 2018A03031042, and Guangzhou Science and Technology Planning Project, No. 201804010138, and the Program for Changjiang Scholars and Innovative Research Team in University, No. IRT_16R37. The funders had no role in study design, data collection and analysis, decision to publish, or preparation of the manuscript.

### Grant Disclosures

The following grant information was disclosed by the authors:
Natural Science Foundation of China: 81772651.
Natural Science Foundation of Guangdong Province: 2018A03031042.
Guangzhou Science and Technology Planning Project: 201804010138.
Program for Changjiang Scholars and Innovative Research Team in University: IRT_16R37.

### Competing Interests

The authors declare that they have no competing interests.

## Author Contributions

- Yixin Tian conceived and designed the experiments, performed the experiments, analyzed the data, prepared figures and/or tables, authored or reviewed drafts of the paper, and approved the final draft.
- Yiquan Ke conceived and designed the experiments, performed the experiments, prepared figures and/or tables, and approved the final draft.
- Yanxia Ma conceived and designed the experiments, performed the experiments, analyzed the data, prepared figures and/or tables, authored or reviewed drafts of the paper, and approved the final draft.

## Data Availability

The gene expression profiles were downloaded from The Chinese Glioma Genome Atlas (http://www.cgga.org.cn/download.jsp) Part B, DataSet ID: mRNAseq_693.

## Supplemental Information

Supplemental information for this article can be found online at http://dx.doi.org/10.7717/peerj.9038#supplemental-information.

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
