# Peer review of "High expression of stromal signatures correlated with macrophage infiltration, angiogenesis and poor prognosis in glioma microenvironment"

_PeerJ, doi:10.7717/peerj.9038_

## Round 0.1 · original submission · Major Revisions

The reviewers have recommended publication pending major revisions. Therefore, I invite you to respond to the reviewers' comments at the bottom of this letter and revise your manuscript accordingly.

Reviewer 1 ·

Basic reporting

no comment

Experimental design

no comment

Validity of the findings

no comment

Additional comments

High expression of stromal signatures correlated macrophage infiltration, angiogenesis and poor prognosis in glioma microenvironment is well structured and well written.
I have made some considerations and would like the authors to review, read and restructure the following suggested points:
1.How about the protein level and the RNA transcript level of HSPG2, FOXF1, KDR, COL3A1, SRPX2 in the clinic samples?
2. question 1 should also be discussed in the discussion section.

Reviewer 2 ·

Basic reporting

The topic is of clinical importance. However, the methods are not solid enough to draw the conclusion. The major weakness is lacking of tissue examination for glioma infiltrating macrophages. In general, the scientific merit is not enough to be published in the current content.

Experimental design

1. The clinical data from CGGA and TCGA may need to be stratified according to current standard prognostic factors for further analysis of stromal signature.

2. No tissue examination for validation of macrophage infiltration predicted by CIBERSORT.

Validity of the findings

The macrophage infiltration. especially for the M2 infiltration needs to be validated.

Additional comments

None.

Reviewer 3 ·

Basic reporting

no comment

Experimental design

no comment

Validity of the findings

no comments

Additional comments

I congratulate the authors for using public available data sets to conduct highly innovative and impactful research on gliomas. In the paper, the authors provide the websites where the CGGA and TCGA data sets are downloaded. However, it would be more more useful to the general readers if the authors can provide more descriptive detailed information about the data used in the paper, particularly for those downloaded from the Chinese CGGA website. More detailed descriptive metadata identfiers are useful to future general readers. The study design is well thought through and comments and discussion are useful and insightful. My suggestion is to provide more details on the statistical analyses. Examples include (1) provide more details on the model the at generated the classifier on CGGA and based on the classifier the ROC and AUC is calculated. Also, the classifier used in the TCGA datset, is there any model fitting used again, or just using the same classifier obtained from the CGGA dataset? If the same approach is used with TCGA as training and CGGA as validation, are the AUCs similar? (2) Are Kaplan-Meier curves to compare high versus low groups all based on median cut off? If not, specify the method used; (3) Are the p-values comparing the Kaplan-Meimer curves log-rank tests? Improve the description of the statistical analyses used can help the future readers; a last minor point, (4) the authors made novel contribution on impact of cells at tumor microenvironment on giloma prognosis, there is a recent publication by Sun X. et al (2019, Journal of Translational Medicine, vol 17, article 159) which discussed a macrophages-related gene signature predictive of survival of glioma patients also using the CGGA and TCGA datasets. It might be of some interest for the authors to comment on whether there is any connections between the signatures.

·

Basic reporting

1. Citation for CGGA and TCGA are missing.
2. The tool for applying the GSEA is not mentioned
3. R, R packages (survival and pROC) GO, KEGG, STRING and Cytoscape are not cited properly.
4. Method and cutoff for dividing samples into high/low infiltration (of immune cells) and expression (of the five genes) are not mentioned.
5. Figure 4A-D. The figures show a significance value (-log10 FDR) but no estimate (like, a fraction of differentially expressed features in each term) which might be useful to show. Arguably, the raw p-values should be used since these are the per-gene quantity unlike the FDR values.
6. The generation of PPI network and the stromal gene signature are not described in enough details in the methods section. What exactly was the use for MSigDB? Please, clarify.
7. Tables describing the clinical characteristics of the CGGA and TCGA sets might be useful to add to the manuscript.
8. Figure 6G is not reference in the text.

Experimental design

The selection of 102 core genes and five stromal genes from the functional 3 gene sets wasn’t fully described or justified. This was done apparently based on their correlation with angiogenesis (Line 139). Arguably, considering a larger number of genes could improve the predictive value of the signature. Further validation of the stromal signature in the TCGA dataset may be required. Other than the ROC curve, the expression of the individual genes and their association might be useful to study.

Validity of the findings

The five stromal gene signature correlates with the tumor grade, but not the histology! Similarly, the signature correlates with the grade but not the response to treatment (Radio or chemo). Please, discuss.

---

## Round 0.2 · accepted · Accept

I am writing to inform you that your manuscript - High expression of stromal signatures correlated with macrophage infiltration, angiogenesis and poor prognosis in glioma microenvironment - has been Accepted for publication

·

Basic reporting

no comment

Experimental design

no comment

Validity of the findings

no comment

Additional comments

The revised manuscript addressed most of the issues raised in the review. In particular
1. Missing references and methodological details were added
2. The method for selecting core genes and the signature was clarified. Moreover, the expression of the five genes was also added to the manuscript.
3. The correlation between the proposed signature and the histology/treatment was further discussed. These correlations suggests that the proposed signature has a potential for clinical applications.